# Effect of Face Masks on Blood Saturation, Heart Rate, and Well-Being Indicators in Health Care Providers Working in Specialized COVID-19 Center

**DOI:** 10.3390/ijerph19031397

**Published:** 2022-01-26

**Authors:** Izabela Wojtasz, Szczepan Cofta, Paweł Czudaj, Krystyna Jaracz, Radosław Kaźmierski

**Affiliations:** 1Department for Neurology with Stroke Unit, L. Bierkowski Hospital, 60-631 Poznan, Poland; iza.wojtasz@gmail.com; 2Institute of Health Sciences, Collegium Medicum, University of Zielona Gora, 65-417 Zielona Gora, Poland; 3Department of Pulmonology, Allergology and Respiratory Oncology, Poznan University of Medical Sciences, 60-569 Poznan, Poland; scofta@ump.edu.pl; 4Faculty of Health Sciences, Poznan University of Medical Sciences, 60-179 Poznan, Poland; paulczudaj@gmail.com; 5Department of Neurological Nursing, Faculty of Health Sciences, Poznan University of Medical Sciences, 60-179 Poznan, Poland; jaracz@ump.edu.pl; 6Department of Neurology, Collegium Medicum, University of Zielona Gora, 65-046 Zielona Gora, Poland; 7Department of Neurology, Poznan University of Medical Sciences, 61-701 Poznan, Poland

**Keywords:** SARS-CoV-2, COVID-19, FFP2 respirator, oxygen saturation, personal protective equipment, face masks, heart rate, fatigue

## Abstract

This study aims to investigate whether wearing face masks (filtering facepieces, FFP class 2) with personal protective equipment (FPP2/PPE), while working a 12-h shift in a COVID-19 referral center, affects the blood saturation, heart rate (HR), and well-being of health care providers (HCPs). The study included a group of 37 HCPs. To perform continuous recordings of the SpO_2_ and heart rate (HR) in real time, we used a Nellcor PM10N (Covidien, Mansfield, MA, USA) portable monitoring system. SpO_2_, HR, and HCP well-being scales were measured during two 3-h shifts, while HCPs worked during a 12-h period. Additionally, each subject completed a questionnaire concerning their well-being. The difference in the SpO_2_ level between the 1st and 2nd working shifts while wearing an FFP2/PPE was small, with a median decrease in SpO_2_ of −1%. The scales of the well-being indicators increased within the shift. They were mainly fatigue and thirst with median scores of 2 out of 6 (range 0–4). We assume that during a 12-h period, a work scheme that consists of a 3-h shift in FFP2/PPE and a 3-h rest period (working without FPP2/PPE) is a reliable and safe solution for HCPs working in specialized COVID-19 referral hospitals.

## 1. Introduction

Health care providers (HCPs) who take care of COVID-19 patients are subject to direct exposure to the SARS-CoV-2 virus. During the COVID-19 pandemic, specialized COVID-19 referral centers are the main and usually last line of defense against COVID-19. In such hospitals, HCPs face an influx of critically ill COVID-19 patients and require heightened personal protective equipment (PPE). Therefore, HCPs in specialized COVID-19 referral hospitals must wear surgical face masks (filtering facepieces, FFPs), eye protection (safety spectacles or goggles), and special gowns that constitute PPE.

Recently, several studies have confirmed the effectiveness of face masks in the COVID-19 pandemic in terms of protecting both HCPs and their patients [1,2,3,4,5,6,7]. Wearing face masks reduces the risk of COVID-19 infections by 70 to 79% in different social, professional, and private settings [8,9,10,11,12].

The FFP 3 respirators have a filter efficiency of 99% inward, and leakage of 2%, as well as have a reliable resistance against patients with a SARS-CoV-2 infection [12,13,14,15]. N95 respirator is a “respiratory protective device designed to achieve a very close facial fit and very efficient filtration of airborne particles”. The U.S Food and Drug Administration (FDA) states that N95 respirators block at least 95% (and FFP2 blocks 94% particles), and FFP1 respirators demonstrated only about 80% bacterial filtration efficiency, which means that the capacity to filter 0.3 μm test particles is 94% for FFP2 and 99% for FFP3 respirators [13,14,15,16,17].

There are some disagreements concerning the health and safety of wearing face masks. One hypothesis assumes that masks can increase resistance to inspiration and respiration, thereby decreasing the efficacy of breathing [18].

Surprisingly, few reports have shown the influence of wearing FFP class 2 (FFP2) respirators on blood saturation, heart rate (HR), or the well-being of HCPs working in COVID-19 departments during 12-h shifts [19,20].

This study aims to investigate whether wearing FFP2 with PPE (FPP2/PPE), while working 12-h shifts in a specialized COVID-19 referral center, affects the blood saturation, HR, and well-being of HCPs.

## 2. Materials and Methods

### 2.1. Study Time and Participants

The study was performed in May 2021 in the Specialized COVID-19 Referral Center of Poznan University of Medical Sciences situated in The Temporary Hospital at the Poznan International Fair market hall.

The study included a group of 37 HCPs (including 27 females (73%) and ten males), with a median age of 22 years, with interquartile ranges (IQRs) of the participants (21–24) and a range of 21–50 (mean (SD) 25.5 years of age (±7.87)).

The study group consisted of registered nurses (RN) (*n* = 6, with 6, 8, 12, 14, 16, and 17 years of professional experience, respectively), nursing students (second and third years of study) (*n* = 24), medical students (fifth year) (*n* = 5), and 2 attendings (MD with 14 and 18 years of professional experience). All participants were Caucasian.

All participants had valid medical examinations and were allowed to work as recognized by an occupational doctor, and all were vaccinated twice with the COVID-19 vaccine (Pfizer/BioNTech, New York, NY, USA). Persons with a history of pulmonary disease, anemia, vascular diseases, and other diseases that could limit respiratory function were excluded from the study.

### 2.2. Equipment

We used FFP2 masks (FFP2 RD, CEI EN149:2001+1A:2009, 2020-1XG (KN95), without exhalation valve) with gowns (Disposal Medical Integrated Protective Coverall, RAEX), and goggles. Additionally, we asked the participants not to use any fingernail polish.

Blood saturation (SpO_2_) and HR were measured in real time for the duration of the 3-h working shift with a pulse oximeter (Nellcor-Portable SpO_2_ Patient Monitoring System, PM10N, Covidien, Mansfield, MA, USA). The single-use SpO_2_ sensor (Nellcor, Neonatal-Adult SpO_2_ Sensor, MAXIN, Covidien, Mansfield, MA, USA) was kept under the surgical glove on the fifth finger of the nondominant hand and connected under the gown with a PM10N device.

### 2.3. Research Design

The SpO_2_, HR, and HCPs well-being scale were measured during two 3-h shifts, while HCPs worked a 12-h period (3 h of work, followed by 3 h of rest, 3 h of work, 3 h of rest) where the “rest” shift means that HCPs performed professional activities (out of the “red zone”) that did not require wearing FFP2/PPE, but instead, only FFP1 surgical masks and lab coats were worn (See Appendix A for details).

In special extraordinary situations, due to a temporary lack of staff, some RNs (*n* = 4) were asked to stay in the ward longer, i.e., a 24-h work period (in such situations, we additionally recorded parameters during shift 3 and shift 4, i.e., between 12 and 15 h and between 18 and 21 h of work).

Additionally, every 30 min during the three-hour shift, each subject completed a questionnaire concerning their well-being, with a score ranging from 0 to 6 points for symptoms including headaches, shortness of breath, perspiration, fatigue, and thirst. 

The score scale is shown below:

0 points—No symptoms;

1 point—Small symptoms, acceptable, non-disruptive work;

2 points—Mediocre symptoms, non-disruptive work;

3 points—Moderate symptoms, with slight work disruption;

4 points—Burdensome working conditions that disrupt work;

5 points—Severe symptoms that considerably disrupt work;

6 points—Symptoms forced the work to be broken off (an exact interruption time should be given).

SpO_2_ and HR were measured continuously (over 752,000 measurements were collected from 37 participants).

### 2.4. Statistical Methods

For the statistical analysis, the data were anonymized (except for information about participants’ sex and age). Additionally, the data have been stored on electronic media (back-up hard drive) without network access. Descriptive statistics with a division into sex and age groups were calculated. Table 1 reports the number of measurements *(N)* and within a single shift median and interquartile range due to the need to implement nonparametric tests and to compare the two shifts of the SpO_2_ and HR results with regard to sex and age.

Comparisons of differences between all shifts in 30-min intervals for each participant’s SpO_2_ and HR mean ranks were performed using the nonparametric Kruskal–Wallis test (Table 2). A comparison of differences was performed between shifts 1 and 2 for SpO_2_ results in 30-min intervals, for each participant. SpO_2_ mean ranks were performed using the Wilcoxon (Mann–Whitney U) test (Table 3). Spearman’s rank correlation coefficient of monotonic association between SpO_2_ and variables describing patient complaints, along with their statistical significance assessed in the study, are shown in Table 4. Figure 1A–D present median and interquartile rate (IQR) graphs of HCPs well-being in terms of shortness of breath, fatigue, thirst, and perspiration. In each case, for a given 30-min time interval, medians and IQR were compared between shift 1 and shift 2.

**Table 1 ijerph-19-01397-t001:** Characteristics of SpO_2_ and HR results by sex and age for shift 1 and shift 2.

Variable	SpO_2_ [%, Median, IQR **]		HR [bpm, Median, IQR **]	
*HCP* *characteristics*	*Shift 1*	*Shift 2*	*Shift 1*	*Shift 2*
*Females (* N = 268,996)*	*97 (96–98)*	*97 (96–98)*	*103 (92–113)*	*99 (87–110)*
*Males* *(* N = 100,217)*	*96 (95–96)*	*96 (95–97)*	*104 (89–115)*	*99 (82–111)*
*<35 years* *(* N = 326,115)*	*98 (97–98)*	*97 (97–98)*	*105 (93–115)*	*102 (89–112)*
*≥35 years* *(* N = 43,098)*	*97* *(96–98)*	*96 (96–97)*	*92 (82–100)*	*85 (75–91)*

* *N*—number of measurements, ** IQR—interquartile ranges.

**Table 2 ijerph-19-01397-t002:** The median (IQR) of the total and for shifts 1 and 2, respectively, in 30-min intervals for SpO_2_ and heart rate (HR).

Variable	30-Min Intervals (Median, IQR)						Test
	0–30	31–60	61–90	91–120	121–150	151–180	
SpO_2_ [%] Total	97 (96–98)	97 (96–98)	97 (96–98)	97 (96–98)	97 (96–98)	97 (96–98)	*Kruskal–Wallis rank sum test p < 0.001*
Shift 1st	97 (96 99)	97 (96–97)	97 (96–97)	97 (96–97)	96 (96–97)	96 (96–97)	
Shift 2nd	97 (96 98)	97 (96–98)	97 (96–98)	97 (96–98)	97 (96–98)	97 (96–98)	
HR (bpm) Total	104 (89–113)	103 (88–112)	101 (85–111)	97 (84–109)	98 (82–111)	96 (81–109)	
Shift 1st	105 (92–114)	105 (95–114)	103 (91–113)	102 (90–112)	103 (90–114)	100 (88–112)	
Shift 2nd	104 (92–114)	102 (89–113)	100 (84–110)	96 (85–108)	97 (83–109)	95 (83–108)	

**Table 3 ijerph-19-01397-t003:** Comparison of differences (*p*-value) between shifts 1 and 2 and between shifts 3 and 4. SpO_2_ results in 30-min intervals for each participant’s SpO_2_ mean level calculation.

Variable *	*p*-Value						Test
SpO_2_	0–30 min	31–60 min	61–90 min	91–120 min	121–150 min	151–180 min	
Shift 1 vs. Shift 2	0.823	**0.009 ****	**0.011 ****	**0.021 ****	0.054	0.197	Wilcoxon
Shift 3 vs. Shift 4	0.25	0.25	0.25	0.75	0.25	0.5	Wilcoxon

* There were no significant differences between the shifts for the heart rate. ** Statistical significance is marked by bold values.

**Table 4 ijerph-19-01397-t004:** Spearman’s correlation coefficients between SpO_2_ and the variables describing HCPs well-being, assessed in the study (*n* = 37, HCPs).

Variables (HCPs, *n* = 37)	Number of HCPs with Positive Variables History, *n*, (%)		Correlations Coefficients between SpO_2_ and the Sum of Variables	*p*-Value
	1st shift	2nd Shift		
Shortness of breath	24 (65%)	26 (70%)	−0.121	<0.001
Fatigue	28 (76%)	32 (85%)	−0.057	<0.001
Thirst	34 (92%)	35 (94%)	−0.142	<0.001
Headache	8 (22%)	9 (26%)	−0.006	<0.001
Perspiration	28 (76%)	29 (78%)	−0.127	<0.001

**Figure 1 ijerph-19-01397-f001:**
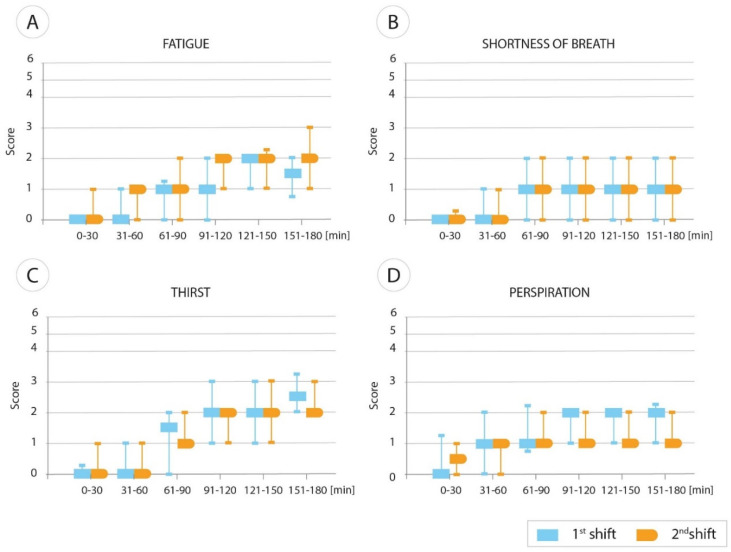
(**A**–**D**). Graphs showing HCP well-being in terms of fatigue (**A**), shortness of breath (**B**), thirst (**C**), and perspiration (**D**) in the 1st and 2nd shifts (median, IQR).

## 3. Results

The means for all HCPs and all shifts for SpO_2_ were 97% (IQR 96–98) and for HR 101 (IQR 88–112) beats per minute (bpm).

Characteristics of SpO_2_ and HR results by sex and age for shift 1 and shift 2 are depicted in Table 1.

In the group working longer periods, up to 24 h, we also analyzed shifts 3 and 4, i.e., shifts between 12 to 15 and 18 to 21 h of work. There were no clinically significant differences in the second 12-h work period or between the third and fourth shifts in terms of SpO_2_ and HR.

For shifts 3 and 4, the recorded results (median, IQR) were as follows: 98% (97–98) and 97% (96–98) for SpO_2_ and 75 bpm (68–82) and 69 bpm (61–78) for HR for a given shift, respectively.

For all four shifts that included HCPs staying for a 24-h period, there were no clinically significant changes in terms of SpO_2_ and HR.

Comparisons of differences between all shifts in 30-min intervals for each participant’s SpO_2_ and HR mean ranks are shown in Table 2.

There were significant differences in participants’ SpO_2_ between shifts 1 and 2 in the period between 31 and 120 min (Table 3).

However, there were no significant differences between men and women or between older and younger (<35 years) HCPs in terms of SpO_2_ and HR.

We also assessed differences between the first 10 min (minutes 1–10) of work in FFP2/PPE and the last 10 min (181–190) of each shift. Pooled results showed small differences between these periods (a decline in SpO_2_ from 98% (96–99) to 96% (96–97), and for HR from 103 bpm (91–115) to 102 bpm (90–113)).

We found negative and significant, but again rather weak correlations, between SpO_2_ levels and HPCs’ well-being factor scores (for details, see Table 4).

HCP well-being indicators presented as medians of the pooled score during a 3-h shift in 30-min intervals are depicted in Figure 1. In the figure, each pair of boxes in different colors shows a comparison between the 1st shift and 2nd shift.

The number of all HCP well-being negative indicators increased in the 2nd shift compared to the 1st shift (Table 4). The data depicted in Table 4 and Figure 1 show that although most HCPs indicated the presence of the well-being factors analyzed in the surveys, the scores were usually relatively low (and most often, the scores were 1 or 2 out of 6).

Headaches are not depicted in Figure 1 because there were a low number of positive scores, and the median was “0” for all 30-min periods (range 0–4).

## 4. Discussion

Although HCPs have been working long shifts throughout the whole pandemic period that has now exceeded two years, the question of how wearing masks influences SpO_2_ is ambiguous in the literature. Different kinds of masks were tested under different experimental conditions [19,20,21,22,23,24,25]. Additionally, PPE gowns could impair cutaneous respiration, causing an increase in sweating and a feeling of discomfort.

Recently, Brooks and Butler stated that concerns about reduced oxygen saturation when wearing a mask have not yet been supported by available data. In fact, there are still very limited data concerning the health safety and comfort of HCPs working for long periods in FFP2/PPE in contagion wards [11].

Therefore, our work aims to provide real-world data from the Temporary COVID-19 Hospital under pandemic third-wave circumstances, and we hope to shed some light on this problem.

We found statistically significant differences between shifts 1 and 2 in terms of SpO_2_ and HR; nevertheless, the absolute median difference between the start and finish of the shifts did not exceed 1–2% for SpO_2_. Even those working a very long 24-h period showed a maximum drop of 2% and usually only a 1% drop in SpO_2_ between the 1st, 2nd, and 3rd, and 4th shifts (decline from 98% (IQR 96–99) to 97% (IQR 96–97)). Such small fluctuations, given the correct starting parameter, should not have any clinical meaning and should not have an impact on the state of healthy HCPs. The changes between shifts 1 and 2, in terms of SpO_2_, were significant in the period between 31 and 120 min, especially between 91–120 min; however, again, the difference did not exceed 1% of SpO_2_.

In Table 3, the correlations were also weak due to the calculation methods where the means of all records of each HCP were taken into consideration. In contrast, in Table 2, a very large number of records results in statistical significance despite a lack of clinically significant changes.

Regarding shortness of breath, it is worth noting that in case of difficulty breathing, it should result in a shortage of inhaled oxygen, which in turn should stimulate the sympathetic nervous system and an increase HR. However, we observed a decrease in HR during consecutive shifts. We did not observe any increase in HR during any shift, again demonstrating that there were no perceptible anoxia episodes in the HCPs.

Other studies concerning the influence of wearing FFP2 or FFP3/PPE on SpO_2_ and HR deal primarily with changes in SpO_2_ and HR during exercise [23] or surgical operations [20,22]. In the above cited papers, SpO_2_ and HR were measured twice: once at the beginning and once after surgery; in contrast to our study, where all measurements were performed continuously, and we recorded over 10,150 records per HCP per shift.

Recently, Fantin performed an arterial blood gas analysis in a 30-year-old female resident, which was carried out during a long 13-h day shift in a COVID-19 Intensive Care Unit. This study found that while wearing an FFP3 mask, all measured values were within normal ranges, although she had an increase in the partial pressure of carbon dioxide (PaCO_2_), from 29.3 to 36.7 mmHg, showing a continuous decrease during the shift in the partial pressure of oxygen (PaO_2_), from 102 to 80.8 mmHg. As the study consisted of just one young volunteer, no further conclusions can be drawn from it [25].

In another study, the authors stated that subjects with COPD did not exhibit major physiologic changes in gas exchange measurements after a 6-min walk test using a surgical mask, particularly in CO_2_ retention [19].

Regarding nonmedical masks, Chan et al. recently showed results that do not support claims that wearing nonmedical face masks in community settings are unsafe [24].

We observed some increases in well-being markers over the duration of each shift; however, there were no significant differences between shifts 1 and 2. In fact, in both shifts, some parameters increased over time (Figure 1A–D). The most evident increase is in thirst and fatigue; however, the symptoms were mild, and in none of the observed parameters did the scale exceed a median score of 2 (out of 6). For fatigue and thirst, the score scales ranged from 0 to 4 and 0 to 2, respectively. We also demonstrated rather weak Spearman’s correlation coefficients between SpO_2_ and the remaining variables assessed in the study (Table 4).

Regarding the number of HCPs with a positive history of well-being factors, our results differed from those of other authors. Barycka et al. demonstrated that among HCPs wearing FFP2 (N95) respirators, headaches were reported in 1.3% of participants, and difficulty breathing was reported in 19.4% vs. 22% and 65% (in the first shift) in our study, respectively. Such differences could be due to methodological differences between the studies [26]. On the other hand, another study analyzed the effects of wearing N95 facemasks on subjective sensation, such as breathing resistance, and showed a maximal rating of 5 out of 10 after 90 min of testing, which is generally in agreement with our results [21].

The different kind of sensations connected with mask wearing attracted the increasing attention of researchers. Kanzow et al. showed that the use of face masks increased the perception of dry mouth and halitosis [27]. Other recent study results suggest that N95 (FFP2) respirators are able to induce an increased facial skin temperature, greater discomfort, and lower wearing adherence when compared to the medical surgical masks [28]. 

Physiologically, normal SpO_2_ readings usually range from 95% to 100%. Values under 90% are considered low, and values of 92% are sufficient to predict adequate oxygenation in Caucasian persons (a saturation of 95% is usually required for persons of African descent) [29,30,31].

Therefore, the obtained results do not exceed the level that is considered safe. However, we should bear in mind that HCPs have been working in COVID centers for the entire pandemic period, which has now exceeded two years. Therefore, to avoid burn-out and chronic fatigue syndromes, we should seek to ensure as much comfort as possible while working [32,33,34]. 

## 5. Conclusions

In this real-world study, we demonstrated that the parameters changed slightly between the 1st and 2nd shifts, and although the changes were statistically significant due to the large number of records, they did not exceed the norm. It is unlikely that such a SpO_2_ decrease of 1–2% (within the range 96–98%) could influence HCP health.

Despite small physiological changes, we found that most HCPs demonstrated mild to moderate discomfort while working with FFP2/PPE, and they complained mostly of fatigue and thirst. We did not find any differences in well-being parameters between sex or age; additionally, there was weak correlation between the SpO_2_ and the well-being factors used in this study.

Finally, we would like to highlight that during a 12-h shift, with a working scheme consisting of a 3-h shift in FFP2/PPE and 3 h of rest, that is, working without FPP2/PPE (wearing only in FFP1 and lab coat), is a reliable and safe solution for HCPs working in specialized COVID-19 referral hospitals.

## Data Availability

The data presented in this study are available on request from the corresponding author. The data are not publicly available due to ethical and privacy issues.

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
