# Peer review of "Effect of Face Masks on Blood Saturation, Heart Rate, and Well-Being Indicators in Health Care Providers Working in Specialized COVID-19 Center"

_ijerph, 2022, doi:10.3390/ijerph19031397_

Round 1
Reviewer 1 Report
The Introduction section needs to be improved with description of the technical features of equipment for the reader's convenience, what does it mean filtering facepieces, FFP 16 class 2, FPP2/PPE etc, if they are adeguate for the specific contagious and which kind of protection they assure.
Please remove non scientific information like: One concern was raised by young doctors/residents in a hospital in Florida—in terms of safety. Some of them became angry as mask discomfort has been attributed to rebreathing CO2 and to 49 hypoxemia, with some even “considering that masks are lethal” [13].
Please describe the experimental plan with a schematic figure that allows to understand the timing of data collection (morning, night etc) with respect to the entire working period and inset also the scheme of the working week (including days off of work for example).
Please and state the difference between the working habit/rules of the partecipants (if there were) with respect to the pandemic period.
Please state the years of working in the specific role and the qualification (nurse, doctor, virologist, aneshesist...).
Please state which well-being test has been performed, is it validated ? For example what does it mean fatigue? As you know there are several level of fatigue.
Do the authors have other specific data for "stress" or for autonomic nervous system status? Some hormonal level or specific scores.
Please describe the meaning of "thirst", do the participants were obliged not to drink? or do they did not drink for too much work to do. How about their de-hydration ?
Please state the methods also for data on shortness of breath, perspiration.
Please, correct typos like underlined "scrub" (line 88) or space in Tables
Author Response
We want to thank the Reviewers and Editors and for the effort to review our paper and their valuable comments.
Open Review 1
- English language and style
( ) Extensive editing of English language and style required
(x) Moderate English changes required
Amendment:
The paper was edited for proper English language, grammar, punctuation, spelling, and overall style by one or more of the highly qualified native English speaking editors at American Journal Experts (AJE)
The verification code is A689-44F2-2D15-5411-5E26.
- The Introduction section needs to be improved with description of the technical features of equipment for the reader's convenience, what does it mean filtering facepieces, FFP 16 class 2, FPP2/PPE etc, if they are adeguate for the specific contagious and which kind of protection they assure.
Amendment.:
We added a description of the technical features of equipment (the changes were marked in red at copy with highlighted changes ).
The FFP 3 respirators have a filter efficiency of 99% inward and leakage of 2%. The N95 respirator is a “respiratory protective device designed to achieve a very close facial fit and very efficient filtration of airborne particles”. The U.S Food and Drug Administration (FDA) describes N95 respirators block at least 95% (and FFP2 blocks 94%) 0.3 μm particles, and FFP1 respirators demonstrated only about 80% filtration efficiency. Which means that , the capacity to filter 0.3 μm test particles is 94% for FFP2 and 99% for FFP3 respirators.
We also added new positions to bibliography:
- Centers for Disease Control And Prevention (CDC). Updated Healthcare Infection Prevention and Control Recommendations in Response to COVID-19. Vaccination http://www.cdc.gov (2021).
- Balty, I. Appareils de protection respiratoire et métiers de la santé. Fiche pratique de sécurité. ED 105. inrs.fr. INRS. April 2020. https://www.esst-inrs.fr/3rb/ressources/ed105.pdf (2020)
- Keeley, L. PPE and COVID-19: what you need to know. The Clinical Services Journal 12, 7-12 (2020)
- Howard, J.; Huang, A.; Li, Z.; Fischer, J.C.; Zänker, K.; van Griensven, M.; Schneider, M.; Kindgen-Milles, D.; Knoefel, W.T.; Lichtenberg, A. et al. An evidence review of face masks against COVID-19. Proc. Natl. Acad. Sci. USA. 118, e201456411. (2021)
- National Institute for Occupational Safety and Health (NIOSH). Use of Respirators and Surgical Masks for Protection against Healthcare Hazards [Internet]; updated 19 November 2018; NIOSH: Washington, DC, USA, 2018. Available online: https://www.
3) Please remove non scientific information like: One concern was raised by young doctors/residents in a hospital in Florida—in terms of safety. Some of them became angry as mask discomfort has been attributed to rebreathing CO2 and to 49 hypoxemia, with some even “considering that masks are lethal” [13].
Amendment: It was removed
- Please describe the experimental plan with a schematic figure that allows to understand the timing of data collection (morning, night etc) with respect to the entire working period and inset also the scheme of the working week (including days off of work for example).
Because it takes up much space, the detailed scheme about the timing of data collection (morning, night etc) with respect to the entire the study period and with the scheme of the working week (including days off of work) were are given in the supplementary materials
where
- Fig 1. Depicted all participants shifts in in the form of duty roster (divided into groups by sex and by professional qualification)
- Fig 2. Showing example of the week's schedule within the whole study period for representatives of all professional groups (i.e. nurses, doctors, students of nursing and medicine students)
5) Please and state the difference between the working habit/rules of the participants (if there were) with respect to the pandemic period.
The study was conducted on during the same pandemic period, between 10 and 26 May 2021 (It was indicated in the section 2.1. Study time and participants)
- Please state the years of working in the specific role and the qualification (nurse, doctor, virologist, aneshesist...).
It was indicated in the section 2.1. Study time and participants)
(All above mentioned changes were marked in red at copy with highlighted changes ).
- Please state which well-being test has been performed, is it validated ? For example what does it mean fatigue? As you know there are several level of fatigue. the details about the score scale with intermediate levels were are given in the supplement, and we are showing it below
The details about the score scale with intermediate levels were are given in the text in the Methods section, and we are showing it below
The score scale (ranging from 0 to 6 points) for the mentioned below symptoms: headache, shortness of breath, perspiration, fatigue, and thirst.
Score scale:
0 points - No symptoms
1 point - Small symptoms, acceptable, non-disruptive work
2 points - Mediocre symptoms, non-disruptive work
3 points - Moderate symptoms, with slight work disruption
4 points - Burdensome working conditions that disrupt work
5 points - Severe symptoms that considerably disrupt work
6 points - Symptoms forced the work to be broken off (An exact interruption time should be given)
- Do the authors have other specific data for "stress" or for autonomic nervous system status? Some hormonal level or specific scores.
No, we did not provided any blood analysis specifically for this study.
- Please describe the meaning of "thirst", do the participants were obliged not to drink? or do they did not drink for too much work to do. How about their de-hydration ?
Amendment.
In the COVID-19 ward - after fitting FFP2/PPE there were not possible to took off the respirator or any part of PPE. After finishing 3-hour shift, all HCPs are obligated to pass through the hospital sluice room, and after taking shower and fitting new clothes they were staying in the green zone - till the next shift. So, during the 3-h shift there were no possibility to drink or eat.
- Please state the methods also for data on shortness of breath, perspiration.
Amendment.
It is subjective scale and it could be difficult to introduce objective tools for headache etc… , however similar to our methodology was used in other studies concerning workers well-being, for example:
Li, Y.; Tokura, H.; Guo, Y.P.; Wong, T.; Chung, J.; Newton, E. Effects of wearing N95 and surgical facemasks on heart rate, thermal stress and subjective sensations. Int. Arch. Occup. Environ. Health 2005, 78, 501-509.
Barycka, K et al. Should emergency medical service stuff use respirators with filtered valve in COVID-19 pandemic? Adv. Respir. Med. 2020, 88, 638-639.
Watanabe K, et al. Measuring eudemonic well-being at work: a validation study for the 24-item the University of Tokyo Occupational Mental Health (TOMH) well-being scale among Japanese workers Ind Health. 2020 Apr 2;58(2):107-131. doi: 10.2486/indhealth.2019-0074
- Please, correct typos like underlined "scrub" (line 88) or space in Tables
Amendment.
We corrected scrubs – for “lab coat” (apron)
Reviewer 2 Report
Comment 1:
In the description of Materials and Methods in part 2.1 "Study time and participants" we observe that if it is indicated that authorization has been requested from the Ethics Committee, to carry out the study, from the COVID-19 Specialized Reference Center of the University of Poznan Sciences Medicine (Ref number 384/21), but it is not indicated whether all participants signed informed consent prior to medical research.
Comment 2:
In the description of Materials and Methods in part 2.3 "Research Design", we note that it does not describe where the data obtained from the study have been stored and if it was carried out in a safe environment, and if access to them was only for those responsible for the investigation.
Comment 3:
Line 234 (generally 95% saturation is required for people of African descent). We consider that it should be specified and contrasted with the bibliography, low oxygen saturation values ​​are common in dark-skinned people, being a common problem that oximeters present, and that affects their accuracy, it would be good if you could put a bibliography to the respect.
Comment 4:
Derived from comment 3, it must be specified in Materials and methods in part 2.1 "Study time and participants" if the study participants were all Caucasian people or dark skinned people.
Author Response
We want to thank the Reviewers and Editors and for the effort to review our paper and their valuable comments.
Open Review 2
1) English language and style
(x) Extensive editing of English language and style required
Amendment.:
The paper was edited for proper English language, grammar, punctuation, spelling, and overall style by one or more of the highly qualified native English speaking editors at American Journal Experts (AJE)
The verification code is A689-44F2-2D15-5411-5E26.
Answers for Reviewers
Comments and Suggestions for Authors
Comment 1:
In the description of Materials and Methods in part 2.1 "Study time and participants" we observe that if it is indicated that authorization has been requested from the Ethics Committee, to carry out the study, from the COVID-19 Specialized Reference Center of the University of Poznan Sciences Medicine (Ref number 384/21), but it is not indicated whether all participants signed informed consent prior to medical research.
Amendment.
At the end of the paper under the point „Institutional Review Board Statement” a next statement was added:
Informed Consent Statement:
Informed consent was signed by all subjects involved in the study prior to medical research.
Comment 2:
In the description of Materials and Methods in part 2.3 "Research Design", we note that it does not describe where the data obtained from the study have been stored and if it was carried out in a safe environment, and if access to them was only for those responsible for the investigation.
Amendment.
Apart from the first and senior author, no other investigator did have access to identifiable protected health information. After the data were obtained. The data were anonymized (the names and surnames of participants were removed) and have been subjected to statistical analysis with only information stored about participants’ sex and age. The data have been stored on electronic media (back-up hard-drive) without network access.
We added the comment in the text: For the statistical analysis the data were anonymized (except for information about participants’ sex and age). Additionally, the data have been stored on electronic media (back-up hard-drive) without network access.
Comment 3:
Line 234 (generally 95% saturation is required for people of African descent). We consider that it should be specified and contrasted with the bibliography, low oxygen saturation values ​​are common in dark-skinned people, being a common problem that oximeters present, and that affects their accuracy, it would be good if you could put a bibliography to the respect.
Amendment.
Thank you for raising this important issue. We added two relevant and important positions of bibliography in the references:
[30] Sjoding MW, Dickson RP, Iwashyna TJ, Gay SE, Valley TS. Racial Bias in Pulse Oximetry Measurement. N Engl J Med. 2020 Dec 17;383(25):2477-2478. doi: 10.1056/NEJMc2029240.
[31] Bickler PE, Feiner JR, Severinghaus JW. Effects of skin pigmentation on pulse oximeter accuracy at low saturation. Anesthesiology 2005;102:715–719
Comment 4:
Derived from comment 3, it must be specified in Materials and methods in part 2.1 "Study time and participants" if the study participants were all Caucasian people or dark skinned people.
Amendment.
We stated in part 2.1. “Study time and participants”
“All participants were Caucasian”.
Round 2
Reviewer 1 Report
I would like to thank the authors for the improvements that clarified all my concerns.